# Encountering Parents Who Are Hesitant or Reluctant to Vaccinate Their Children: A Meta-Ethnography

**DOI:** 10.3390/ijerph18147584

**Published:** 2021-07-16

**Authors:** Sara Fernández-Basanta, Manuel Lagoa-Millarengo, María-Jesús Movilla-Fernández

**Affiliations:** 1Research Group GRINCAR, Department of Health Sciences, Faculty of Nursing and Podiatry, University of A Coruña, Naturalista López Seoane s/n, 15471 Ferrol, Spain; maria.jesus.movilla@udc.es; 2Galician Health Service (SERGAS), University Hospital Complex of Ferrol, Av. da Residencia, S/N, 15405 Ferrol, Spain; manuel.lagoam@udc.es

**Keywords:** anti-vaccination movement, health personnel, qualitative research, professional–patient relations, vaccination refusal

## Abstract

(1) Background: Health professionals play an important role in addressing parents who are hesitant or reluctant to immunise their children. Despite the importance of this topic, gaps remain in the literature about these experiences. This meta-ethnography aimed to synthesise the available body of qualitative work about the care experiences of community and hospital health professionals in encounters with parents hesitant or reluctant to vaccinate their children. The aim is to provide key information for the creation of strategies that address vaccine hesitancy or refusal and ensure public trust in vaccination programs, which are required in a pandemic context such as the current one. (2) Methods: Noblit and Hare’s interpretive meta-ethnography of 12 studies was followed. A line of argument synthesis based on a metaphor was developed. (3) Results: The metaphor “The stone that refuses to be sculpted”, accompanied by three themes, symbolises the care experiences of health professionals in their encounters with parents that hesitate or refuse to vaccinate their children. (4) Conclusions: The creation of clearer communication strategies, the establishment of a therapeutic alliance, health literacy and the empowerment of parents are recommended. The incorporation of health professionals in decision making and the strengthening of multidisciplinary teams interacting with such parents are also included.

## 1. Introduction

Vaccines have reduced infant mortality and morbidity from many infectious diseases, such as poliomyelitis, measles, tetanus, whooping cough and tuberculosis. Despite the multiple benefits, an increasing number of parents choose to delay or decline vaccines [1]. Andrew Wakefield’s misreport in 1998 linking the measles-mumps-rubella (MMR) vaccine to autism sparked a global crisis. Currently, the massive online dissemination of unconfirmed information on vaccination poses a serious risk to public health [2]. Rates of vaccine-preventable diseases have increased in many developed and developing countries [3,4]. Vaccination prevents two to three million deaths per year worldwide; however, 1.5 million deaths could be prevented if due vaccinations were applied. In 2019, measles cases increased by 30%, with resurgences in countries that were close to eliminating the disease [5]. Increasing global mobility and, specifically, international travel pose a risk to people who could be protected from preventable diseases through vaccination [6].

Vaccine hesitancy is defined as “a motivational state of being conflicted about or opposed to getting vaccinated” and represents one of the ten leading threats to global health [5,7]. People who hesitate with regard to vaccines are a heterogeneous group, influenced by various social, cultural, political and personal factors that affect decision making. Worldwide, hesitancy about vaccines has contributed to lower childhood vaccination rates, with associated outbreaks of vaccine-preventable diseases [8,9,10]. Parents hesitant or reluctant to immunise their children are part of this group [1,11]. The reasons why people choose not to vaccinate are complex, including complacency, inconvenience in accessing vaccines and lack of trust [5]. According to demographic variables, reluctant or hesitant parents are more likely to be of higher socioeconomic status, college educated, and have four or more children in the household [4,12,13]. Among the ideological aspects of vaccine refusal, concerns regarding safety and effectiveness as well as mistrust of public health officials drive the choice not to vaccinate [14].

Health workers, especially those in communities, are seen as the most trusted advisor on and influencer of parental decision-making [5,12]. They are responsible for responding to this problem through information campaigns for parents and seek to counter misinformation on vaccines. According to the literature, it is important to create interventions with multidisciplinary approaches and to improve the communication skills of professionals and focus on their relationship with parents [2,15]. Multicompetent interventions based on dialogue with parents were the most successful. Every intervention must be adapted to the target population’s context and characteristics. Overall, training based on communication tools for health workers had a positive impact [16].

However, few studies address the care experiences of healthcare professionals responsible for vaccination in their encounters with parents who are reluctant or hesitant with regard to vaccination [17]. These professionals face many obstacles and difficulties. This study fills a research gap due to the need to better understand the experiences of health workers in their encounters with these parents [18]. Meta-ethnography is an appropriate method that generates new, integrated and more complete interpretations of findings [19]. Since it is a topic highly influenced by contextual and cultural factors as well as by the high probability that health professionals will encounter parents from different cultural contexts, this methodology allows a global and comprehensive vision. On the other hand, the global COVID-19 crisis may influence public trust in public health authorities, varying from country to country, depending on the burden on the country’s health and socioeconomic consequences and the intensity of the controversies. In addition to the segment of the population that rejects vaccines, the novelty of the disease and concerns about the safety and efficacy of vaccines spurred reluctancy in many to get vaccinated against COVID-19 [20,21]. Being aware of the state of hesitation or rejection of vaccines is necessary for general public health, especially when fighting the pandemic. Our results may be useful for developing strategies to address vaccine hesitancy and ensure confidence in the vaccination program against COVID-19.

Thus, this meta-ethnographic study aims to synthesise the available body of qualitative work on care experiences of health professionals from community and hospital settings in encounters with parents hesitant or reluctant to vaccinate their children.

## 2. Related Work

Table 1 shows the main findings of existing reviews on parents’ rejection or hesitancy to vaccinate their children.

The literature indicates that the main reasons for non-vaccination or the appearance of doubts lie in the lack of information and distrust regarding the safety of vaccines. Furthermore, there is no solid body of evidence as to how to approach these parents, although the literature indicates that health professionals play a fundamental role. The results of this meta-ethnography provide a diagnosis from the perspective of health professionals, necessary for the design and implementation of interventions aimed at parents hesitant or reluctant to vaccinate their children.

## 3. Materials and Methods

### 3.1. Design

A meta-ethnographic approach was carried out [27]. This is a method that involves the creation of knowledge by translating individual qualitative studies into each other, reinterpreting, and transforming their findings. The goal is to create deeper and more complete interpretations of the primary articles [28]. This study follows the seven synthesis phases described by Noblit and Hare (1988) [27]: (1) getting started, (2) deciding what is relevant to the initial interest, (3) reading the studies, (4) determining how the studies are related, (5) translating the studies into one another, (6) synthesising translations, and (7) expressing the synthesis.

With the aim of improving quality and increasing transparency and comprehensiveness, the preparation of this review followed the guidelines of the eMERGe reporting guide [29] (Appendix A).

### 3.2. Search Strategy

In the first phase of this meta-ethnography, the research problem and objective were defined. The starting research question was focused on the nursing experience, but due to the scarcity of primary articles that exclusively address this experience, the research question was expanded, resulting in “What are the experiences of health professionals in their encounters with parents hesitant or reluctant to vaccinate their children?”. For the elaboration of the research question, the SPIDER tool [30] was used, serving as a framework for the development of search terms.

The search strategy was designed by SFB (Appendix A). This entailed a comprehensive systematic search of the PubMed, Scopus, CINAHL, Web of Science and PsycINFO databases in January 2020, updated in December 2020. The main databases in health sciences were chosen to identify all potentially relevant titles. The search terms included MeSH (Medical Subject Heading), CINAHL descriptors and free terms. These terms were combined with the Boolean operators “OR” and “AND”. Truncation (*) symbols were used to guarantee a broader search. The limits defined in the databases were idiomatic (English, Spanish and Portuguese). Inclusion and exclusion criteria are detailed in Table 2.

### 3.3. Search Outcomes

The Preferred Reporting Items for Systematic-Reviews and Meta-Analyses (PRISMA) flow diagram [31] shows the filtering process that was carried out by MLM (Figure 1). The search in the databases provided 1060 records, of which 527 were eliminated due to duplication. A total of 533 records were screened according to the title and abstract of the articles and based on the inclusion and exclusion criteria. In the full-reading screening, 32 of 44 records were excluded. The reasons for exclusion were due to an incorrect sample (*n* = 17), incorrect phenomenon of interest (*n* = 9), no primary article (*n* = 4), and incorrect methodology (*n* = 2). The final sample was composed of 12 primary articles. The updated search did not provide new primary articles that met the inclusion criteria.

### 3.4. Quality Appraisal

The quality of each primary article was assessed using the Clinical Appraisal Skills Program (CASP) [32] tool. The quality evaluation of each article was carried out by MLM, and a validation meeting between all the authors was held afterwards. The articles were considered to be of high quality with respect to their objectives, designs, analysis and results, providing useful knowledge on the subject. After this quality evaluation, no study was eliminated, since the objective of this meta-ethnography was not to eliminate any article for its methodological weakness, but rather to seek the richness and strength of the articles’ findings. Table 3 shows the results of this evaluation.

### 3.5. Data Abstraction and Synthesis

The data analysis and synthesis were carried out by MLM and SFB, and meetings between all the authors were held at different times in the analysis process. Phase 3 began with a critical reading of the included studies to describe their objective, sample, methodology, data collection method and key results (Table 4).

The analysis continued with the rereading of the articles until the content of the primary articles was familiarized. The extraction of the first-order (participants ’quotations) and second-order (authors’ interpretations) concepts [45] began with the most data-rich article [41]. These concepts were extracted by MLM in Microsoft Word tables, which favoured intra- and inter-study comparison (step 4). In this step, Table 4 was used as context for the comparisons. In step 5, the concepts from the studies were incorporated into one another by analogous (accounts are directly comparable) and refutational (accounts stand in relative opposition to each other) translations. The meaning of concepts and their relations across study accounts were systematically compared to identify the range of these concepts. Translations from step 5 were compared to identify common or overarching concepts and to develop new interpretations, forming new third-order concepts (meta-ethnography authors’ interpretations) [45].

Finally, in step 6, a storyline of the phenomenon was developed, composing the basis for the line of argument synthesis [29,46]. All the authors agreed on the themes and the overarching metaphor.

The Confidence in the Evidence from Reviews of Qualitative research (CERQual) tool [47] was used to show the degree of confidence in the results of the meta-ethnography (Table 5). The evaluation of each review finding was carried out by MLM and SFB, and meetings between all authors were held for consensus.

## 4. Results

### 4.1. Characteristics of the Studies

Twelve primary articles were analysed, focusing on the experiences of health professionals encountering parents who are hesitant about or refrain from vaccinating their children. This research was conducted in Sweden, the Netherlands, Pakistan, Israel, the United States, the United Kingdom, Slovakia, Canada, and Australia. The number of participants in the primary studies ranged from 11 to 58. The sample consisted of health professionals involved in the administration of childhood vaccines, especially paediatricians, general practitioners, nurses, and child vaccination providers. The data collection methods used were in-depth semi-structured interviews and focus groups. Table 4 shows the main characteristics of the studies included.

### 4.2. Synthesis Results

Through reciprocal and refutational synthesis, the line of argument and the metaphor “The stone that refuses to be sculpted” emerged. This metaphor, accompanied by three themes, symbolises the care experiences of health professionals in their encounters with parents who hesitate or refuse to vaccinate their children. In the metaphor, healthcare professionals are represented as sculptors and parents as the stones to be sculpted. The sculpting process is necessary to achieve a common good and a low-risk environment for society. The first theme, stone hardness, symbolizes the degree of deep-rootedness of the parents’ decision not to vaccinate. The decisions of non-vaccination were constructed based on false beliefs about vaccination, cultural and religious context, or lack of information. In addition, health professionals may lack support, resources or training to deal with these parents, which is represented in the theme of lack of modelling tools. In this context, the professionals had to use their own strategies to develop their roles and convince the parents, represented through the rudimentary sculpting theme.

Table 5 shows the results of the findings assessment with the CERQual tool [47]. Stone hardness (resistance to vaccination) showed high confidence, while lack of modelling tools (lack of resources, support and training) and Rudimentary sculpting (using personal strategies) presented moderate confidence. This means that it is likely that they reasonably represent the experiences of health professionals in their encounters with parents who are hesitant or opposed to the vaccination of their children.

#### 4.2.1. Stone Hardness—Resistance to Vaccination

False beliefs about vaccination, based mainly on a lack of information, myths or the influence of local culture and religion, were identified as the main causes of hesitation towards or rejection of vaccination.

There were two main profiles of parents hesitant or reluctant to vaccinate. On the one hand, were those who were possibly over-informed and knew the risks and benefits of childhood vaccination. Specifically, those parents who belonged to associations against childhood immunisation posed an important challenge for health professionals [34,38]. The members of these associations were highly informed and united, which made them stand firm in their decision not to vaccinate, as one nurse describes:

“*I believe most critical parents are highly educated, difficult to drive an argument home to, having an own opinion but not always reading the scientific literature.*”[38]

On the other hand, there were those who were not well-informed and were unaware of the benefits of childhood immunisation programs. These parents were afraid to administer the vaccine because of issues around safety, effectiveness, or number of doses [39,40,42,43]. Specifically, the fear of parents was also shown by generating a false sense of security regarding human papillomavirus (HPV) vaccination [35].

One of the most prominent false beliefs was the relationship between the MMR vaccine and the development of autism and certain intestinal diseases. Despite this theory being discredited, the idea that vaccinations would cause dumbness in their children was widespread socially [36,41]. Furthermore, homeopathy or traditional/alternative medicine was also, for many parents, a more reliable resource [34,38].

“*It takes time to correct misunderstandings when the parents do not know that a study from Denmark showing that children could be at risk of autism after vaccination was fake.*”Nurse; [41]

Religious motives were an important causal factor in vaccine hesitancy, as shown in the study by Khan and Sahibzada (2016) [37]. For example, some religious leaders held the idea that the blood of some animals prohibited by Islam was part of vaccine composition. These beliefs were often linked to the false belief that vaccination could cause infertility in the child, along with a lack of basic knowledge about general aspects of childhood vaccination.

The influence of pharmaceutical companies and the existence of economic interests also contributed to hesitant parents justifying their decision of non-vaccination, since the information brochures were sometimes sponsored by large pharmaceutical companies [33]. The frequent changes in vaccination programs were related to the influence of pharmaceutical companies, generating mistrust among parents [38].

#### 4.2.2. Lack of Modelling Tools—Lack of Resources, Support and Training

Healthcare professionals often lacked the resources, support, and training to deal with hesitancy or rejection of immunisation. This lack of means was represented in the metaphor as the lack of modelling tools.

The lack of training and information on vaccination among health personnel, mostly nurses, made it difficult to counsel hesitant parents [35,38,41]. The health professionals considered that they did not have enough knowledge about the side effects and other aspects of vaccines to deal with these parents, anticipate their possible questions, or know how to answer them with accurate and updated information [35,38]. The nurses often encountered well-informed parents who asked them about some aspect of vaccination that they did not know how to answer [41].

“*We need more information on the vaccine, about the side effects, so that we can anticipate possible questions and reactions. So that we know what it’s about, simply. Vaccinating is something we’re used to, but this is an entirely new preparation.*”Nurse; [35]

On top of this, the constant changes in immunisation programs and their lack of awareness did not help to improve persuasion and provision of information to hesitant parents. Specifically, the alternative vaccination schedules were a source of doubt for many professionals [34].

Lack of time, poor salaries and lack of recognition by public health systems meant that healthcare professionals did not feel supported to carry out their work. This lack of recognition of vaccinating professionals, particularly nurses, was reflected in the lack of decision-making power regarding changes in vaccination programs. Their opinion was hardly recognised or considered, despite the fact that they were the ones who in most cases faced parents opposed to vaccination [35,38,40].

Moreover, some professionals complained about high workload, since it generated stress to some degree, as expressed by a doctor: “*Mainly it was staffing. It’s a lot of work for the staff to… pick up the vaccines, monitor the fridge temperatures… record them to send in to public health, and… with the number of vaccines increasing, it was just increasingly burdensome for the staff*” [40]. In many cases, the administration of the vaccine coincided with other care, such as periodic health check-ups, which took time away from that dedicated to this aspect [38].

#### 4.2.3. Rudimentary Sculpting—Using Personal Strategies

Health professionals felt that it was a matter of professional responsibility and duty to address the care of parents hesitant or reluctant to vaccinate. Metaphorically, this topic is represented as rudimentary sculpting; in the absence of other resources, health professionals had to develop their own strategies to convince and guide these parents.

Some of the health professionals stated that it was essential to be in favour of vaccination to be able to deal with these situations. As one health professional pointed out, “*You cannot do this job if you do not support the NIP* [National Immunisation Program]” [38]. Specifically, the vast majority of health professionals recognised the numerous benefits of HPV vaccination and considered it as one more vaccine within routine childhood vaccination. However, some professionals had doubts regarding the need for this vaccine, negatively impacting its recommendation [40,43,44]. Furthermore, religious beliefs also influenced some professionals not to recommend adherence to vaccination programs, as was the case for Orthodox Protestants [42]. This caused a conflict between their personal beliefs and their professional duties.

The convincing strategies were based on respect for the opinions, values and decision-making power of parents. Avoiding confrontation and favouring a respectful relationship of trust were believed to have a very positive influence on the long-term results obtained [33,38,39,41,44]. Going against parents would impair cooperation with them. In situations in which parents rejected vaccination from the outset, a positive and empathetic relationship was key to address the issue in future encounters [33,34,38,44].

Furthermore, providing information based on scientific evidence was a fundamental measure [42]. Within the information, details about adverse effects should be included, and these should be transmitted little by little allowing the participation of parents. Health professionals reported that the objective of their recommendations was to convey the safety of vaccines and to make parents see that the risks of contracting the diseases that vaccines prevent were much higher than the adverse effects caused by vaccination. There were also professionals in favour of rescheduling visits, delaying vaccinations or implementing alternative calendars [33].

Despite their efforts, health professionals did not always achieve their goal, and eventually some parents ended up refusing the vaccine. This triggered feelings such as anger, indignation, helplessness and ultimately personal failure [33].

“*(…) we do not have many unvaccinated* [children], *I have about three children. And I have, I have paid for it with high blood pressure, because I have perceived it as a personal failure.*”Paediatrician; [33].

## 5. Discussion

From the analysis of the twelve qualitative primary articles, the line of argumentation and the metaphor “The stone that refuses to be sculpted” emerged. This metaphor represents the experiences of healthcare professionals in their encounters with parents who are hesitant or refuse to immunise their children. Health professionals are represented as sculptors, and parents as the stones to be sculpted. The hardness of the stone showed the profiles of the parents that professionals had to deal with. On one hand, those who had doubts or did not have enough information, and on the other hand, those who were over-informed or their position clearly stood on the rejection of vaccination. These behaviours were influenced by false beliefs about vaccination and the cultural and religious context. During the encounters with these parents, health professionals lacked the necessary tools to address the care of these parents. Lack of resources, support, and training were reported. Faced with this scenario, they had to develop their own strategies to convince and guide these parents.

Planned Behaviour Theory [48] was designed to predict and explain human behaviour. The intention of a behaviour is based on three elements: attitude, social norms and behaviour control. Attitude refers to beliefs that cause a positive or negative attitude towards the behaviour; social norms are related to normative beliefs in society that give rise to perceived social pressure; and control of a perceived behaviour refers to the perceived ease or difficulty of carrying out a behaviour. This theory assumes that attitudes and the perception of control over a behaviour are potentially modifiable aspects on which we can act, planning interventions to achieve a change in behavior [49].

Regarding attitudes, our results showed two profiles of parents. Those who were informed about the benefits or who were strongly positioned against vaccination, and those who did not have enough information or reported doubts about vaccines. Among the factors for vaccine hesitation or rejection, the belief that vaccines are unsafe, the influence of religious beliefs, the mistrust generated by the interests of large pharmaceutical companies and a lack of information stood out [33,34,36,37,38,39,40,41,42,43]. This reflects the importance of parental health literacy on aspects related to childhood vaccination programs. Health literacy falls within the field of health communication and is defined as the degree to which people can obtain, process, understand, and communicate the health-related information necessary to make informed health decisions [50]. Inadequate health literacy results in poorer health, decreased adoption of protective behaviours such as immunization, and a lack of understanding and use of health services [51,52].

Therefore, the parental health literacy assessment contributes to acknowledging limitations and strengths, which will provide key information for the development of strategies that allow adherence to vaccination [53]. These strategies are based on less complex communication with parents, where the characteristics of the context and the communicative format also acquire relevance [54,55]. An effective health professional–parent communication can lead to better health outcomes, such as vaccination acceptance [56]. Health knowledge and parental participation in decision-making must be taken into account for communication to be effective [57].

Feelings of control can be based on previous experiences or influenced by information provided by the social environment [48]. Our findings show that the social, cultural and economic context had a strong influence on the hesitation or decision not to vaccinate. In line with the literature, it is highlighted that the conceptions of health and disease are directly related to people’s education and their cultural, religious and ethnic values. In health communication this plays an important role, since the success of any health program or intervention will largely depend on these factors [58].

According to this, nursing occupies a fundamental position in the care of these parents due to the special training in therapeutic communication and proximity to the parents. Most parents identify closely with nurses, recognising them as part of the same socioeconomic status, in addition to perceiving that they are not part of “the medical establishment” [59]. According to Hoekstra and Margolis (2016) [59], interdisciplinary teams and vaccination programs must take advantage of the qualities and attributes of nurses to strengthen compliance with vaccination programs. The COVID-19 pandemic has highlighted the need to build strong and resilient systems to strengthen confidence in vaccines, to address both current and future threats [60].

### Strengths and Limitations

The strength of this research lies in the methodology used. Meta-ethnography is increasingly used in the field of health sciences. This is due to the great potential to provide a higher level of analysis and to generate new research questions as well as theories from the synthesis of several primary articles [19,61]. The results of the review were evaluated with the CERQual tool [47], showing the confidence and applicability of the results in clinical settings, decision making and future research.

In addition, comprehensive searches at two time points were carried out, with the inclusion of articles written in English, Spanish and Portuguese, ensuring a greater reach. The articles included were also evaluated with the CASP checklist [32], confirming their reliability, transparency and relevance. The preparation of the meta-ethnography followed the eMERGE reporting guidance [29], improving the transparency of the research process and therefore providing methodological rigor.

Moreover, the sample included various professional categories from both the hospital and community spheres, providing a broader vision. However, this variety of professional categories can make it difficult to respond to the study objectives of a particular professional category. New empirical research is required to address these limitations.

## 6. Conclusions

The metaphor and line of argument “The stone that refuses to be sculpted” represents the caregiving experiences of health professionals in their encounters with parents who are hesitant or refuse to vaccinate their children. Healthcare professionals are represented as sculptors and parents as the stones to be sculpted. The sculpting process is necessary to achieve a herd immunity and a low-risk environment for society. These professionals encountered two profiles of parents: those who were over-informed or those whose position was strongly against vaccination, and those who were hesitant about vaccination or did not have enough information. These behaviours were influenced by false beliefs about vaccination and the cultural and religious context. In addition, health professionals found themselves in a context that lacked means, support and training. In this scenario and based on a feeling of professional responsibility and duty, the professionals used their own strategies, based on the establishment of a relationship of trust and respect, to increase the rate of adherence to immunisation.

These results expand the body of knowledge of the disciplines related to childhood vaccination and provide key information to help promote a change in clinical practice, such as the creation of clearer communication strategies, the establishment of a therapeutic alliance, health literacy and the empowerment of parents. In addition, the incorporation of health professionals in decision making and the strengthening of the multidisciplinary teams that encounter these parents are highlighted.

## Figures and Tables

**Figure 1 ijerph-18-07584-f001:**
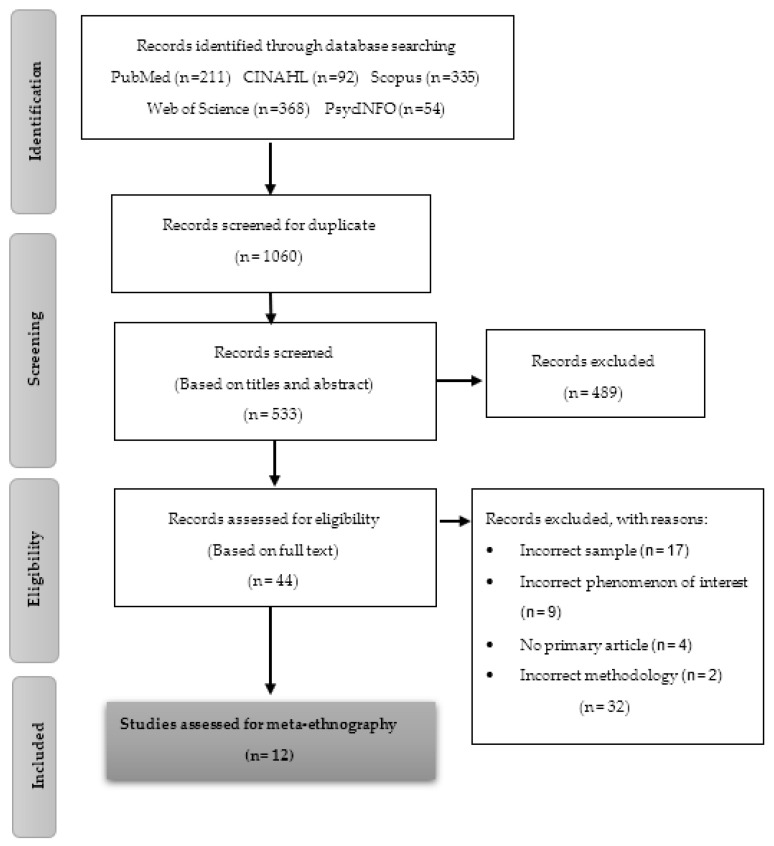
PRISMA flow diagram.

**Table 1 ijerph-18-07584-t001:** Related work contribution.

Aspect	Domain and Papers	Contribution
Reluctant or hesitant parents	Parental reasons for refusal or hesitation of vaccination [22,23]	The main parental reasons for not vaccinating their children are based on religious, personal or ideological reasons, safety concerns, and a desire for more information from healthcare providers. Lack of information and mistrust stand out as the main causes.
Health workers and parents	Interventions [6,16,23,24,25,26]	Educational interventions on risks/benefits, changing the schedule of vaccinations, school-based programs, social norms with culturally tailored messages, and interventions at the physician level were the main interventions, despite the absence of promising results. These works emphasized that educational interventions can be counterproductive. Interventions that include presumptive, announcement language are more likely to be effective than those with participatory and conversational language. Moreover, educational interventions must be carefully tailored according to the target population, their reasons for hesitancy, and the specific context. Apart from the weak evidence regarding the interventions, health professionals reported problems with access to the materials developed to help them in these situations.

**Table 2 ijerph-18-07584-t002:** Inclusion and exclusion criteria.

Inclusion Criteria	Exclusion Criteria
- Articles focusing on the experiences of health professionals responsible for childhood vaccination who encounter parents hesitant or refusing to immunise their children - Original qualitative articles or mixed articles from which the qualitative results could be extracted - Articles published in English, Spanish or Portuguese	Gray literature, discussion or review papers

**Table 3 ijerph-18-07584-t003:** Quality assessment of included studies.

Articles	Questions									
	1	2	3	4	5	6	7	8	9	10
Bašnáková and Hatoková (2017) [33]	✓	✓	✓	✓	-	✗	✓	✓	✓	✓
Berry et al. (2017) [34]	✓	✓	✓	✓	✓	✓	✓	✓	✓	✓
Gottvall et al. (2011) [35]	✓	✓	✓	✓	✓	-	✓	✓	✓	✓
Jama et al. (2019) [36]	✓	✓	✓	✓	✓	✓	✓	✓	✓	✓
Khan and Sahibzada (2016) [37]	✓	✓	✓	✓	✓	-	✓	✓	✓	✓
Mollema et al. (2012) [38]	✓	✓	✓	✓	✓	-	✓	✓	✓	✓
Navin et al. (2020) [39]	✓	✓	✓	✓	✓	-	✓	✓	✓	✓
Omura et al. (2014) [40]	✓	✓	✓	✓	✓	✗	-	✓	-	✓
Rudolfsson and Karlsson (2019) [41]	✓	✓	✓	✓	✓	✓	✓	✓	✓	✓
Ruijs et al. (2012) [42]	✓	✓	✓	✓	✓	✗	✗	✓	✓	✓
Shahbari et al. (2020) [43]	✓	✓	✓	✓	✓	-	✓	✓	✓	✓
Stretch et al. (2009) [44]	✓	✓	✓	✓	✓	-	✓	✓	✓	✓

Abbreviations: ✓ Yes; - Unclear; ✗ No. Critical appraisal questions: (1) Was there a clear statement of the aims of the research? (2) Is the qualitative methodology appropriate? (3) Was the research design appropriate to address the aims of the research? (4) Was the recruitment strategy appropriate? (5) Were the data collected in a way that addressed the research issue? (6) Has the relationship between researcher and participants been adequately considered? (7) Have ethical issues been taken into consideration? (8) Was the data analysis sufficiently rigorous? (9) Is there a clear statement of findings? (10) How valuable is the research?

**Table 4 ijerph-18-07584-t004:** Characteristics of included studies.

Authors (Year) Location	Methods	Aim	Sample	Type of Vaccination	Data Collection Method	Key Findings
Bašnáková and Hatoková (2017) [33] Slovakia	Qualitative study	To identify which communicative strategies Slovak paediatricians implicitly or explicitly choose in order to facilitate parental decisions about vaccination.	15 primary care paediatricians	Mandatory childhood vaccination	In-depth interviews	Paediatricians typically lack formal training in communication with parents, but use a large number of effective communicative strategies that they have acquired during their clinical experience. However, often these decisions are not being made explicitly, and some paediatricians struggle with specific situations and types of parents.
Berry et al. (2017) [34] Australia	Grounded theory	To understand the challenges faced and strategies used when general practitioners and immunising nurses encounter parents who choose not to vaccinate their children.	17 general practitioners and 9 community and practice nurses	Childhood vaccination	In-depth interviews	Providers’ sense of professional identity as health advocates and experts became conflicted in their encounters with vaccine-objecting parents. Providers were dissatisfied when such consultations resulted in a “therapeutic roadblock”, whereby provider–parent communication came to a standstill. There were mixed views about being asked to sign forms exempting parents from vaccinating their children. These ranged from a belief that completing the forms rewarded parents for non-conformity to seeing it as a positive opportunity for engagement. Three common strategies were employed by providers to navigate through these challenges: (1) to explore and inform, (2) to mobilise clinical rapport and (3) to adopt a general principle to first do no harm to the therapeutic relationship.
Gottvall et al. (2011) [35] Sweden	Qualitative study	To investigate school nurses’ perceptions of HPV immunization and their task of administering the vaccine in a planned school-based program in Sweden.	30 school nurses	Human Papillomavirus (HPV) immunization	Focus groups	The school nurses saw the program as a benefit in that the free school-based HPV immunization program could balance out social inequalities. However, they questioned whether this new immunization program should be given priority given their already tight schedule. Some also expressed doubts regarding the effect of the vaccine. It was seen as challenging to obtain informed consent as well as to provide information regarding the vaccine. The nurses were unsure of whether boys and their parents should also be informed about the immunization.
Jama et al. (2019) [36] Sweden	Explorative with inductive qualitative approach	To explore the perceptions, views and experiences of children’s health clinic nurses related to vaccine hesitancy in Rinkeby and Tensta.	11 children’s health clinic Nurses	Measles-mumps-rubella (MMR) vaccination	In-depth interviews	Four themes emerged, namely hesitancy among Somali parents, lack of confidence in the MMR vaccine, loss of confidence in other vaccines due to mistrust of the MMR vaccine, and complacency regarding vaccination in general.
Khan and Sahibzada (2016) [37] Pakistan	Qualitative study	To explore the challenges faced by health workers (HWs) during the polio health campaign.	42 health workers (HWs)	Oral polio vaccine (OPV)	Focus groups	HWs disclosed that public attitude and harsh behaviour towards the HWs and security threats are the two main challenges they face. Common issues hindering parents’ willingness to vaccinate their children against OPV are that OPV is seen as *haram* and not permitted in Islam, it is said to contain the blood of pigs and monkeys, and parents are afraid that it is done to induce sterility among their children. HWs also shared that parents have a strong belief in the conspiracies that are associated with OPV, i.e., the USA and CIA are spying on us and our government is helping them to achieve their agenda. Furthermore, HWs revealed that frequent visits may further strengthen parents’ perceptions and make them more resistant to OPV.
Mollema et al. (2012) [38] Netherlands	Exploratory qualitative study	To examine the factors behind the intentions to recommend current and future vaccinations to parents.	25 child welfare centre nurses and physicians	Non-mandatory childhood immunization	Focus groups	Four main themes emerged, including (1) perceived responsibility: to promote vaccines and discuss pros and cons with parents (although this was usually not done if parents readily accepted the vaccination); (2) attitudes toward the NIP: mainly positive, but doubts remained as to NIP plans to vaccinate against diseases with a low perceived burden; (3) organizational factors: limited time and information can hamper discussions with parents; (4) relationship with parents: crucial and based mainly on communication to establish trust.
Navin et al. (2020) [39] USA	Grounded theory	To attend to the activity and dispositions of the public health staff who provide “waiver education”.	39 local health department staff (37 licensed nurses)	Routine childhood immunizations	Focus groups	Four themes emerged from analysis of the transcripts of these interviews: Participants had (1) complex and nuanced observations and evaluations of parents’ judgments and feelings about vaccines and vaccine education; (2) sympathetic attitudes about alternative vaccine schedules; (3) critical and supportive evaluations of institutional policies and the background political context of immunization education; and (4) consistent commitments to respect parents, affirm their values, and protect their rights.
Omura et al. (2014) [40] Canada	Qualitative study	To explore the experiences of family physicians and paediatricians delivering immunizations, includingperceived barriers and supports.	46 family physicians or general practitioners, 10 paediatricians and 2 residents	Routine childhood immunizations	Focus groups	Physicians highly valued vaccine delivery. Factors facilitating physician-delivered immunizations included strong beliefs in the value of vaccines and having adequate information. Identified barriers included the large time commitment and insufficient communication about program changes, new vaccines, and the adult immunization program in general. Some physicians reported good relationships with local public health, while others reported the opposite experience, and this varied by geographic location.
Rudolfsson and Karlsson (2019) [41] Sweden	Qualitative study	To explore nurses’ experiences of encountering parents who are hesitant about or refrain from vaccinating their child.	12 nurses from 7 child healthcare centres	Childhood vaccination	Individual, semi-structured interviews	Three themes emerged from the interviews: giving room and time for acknowledging parents’ insecurity concerning vaccination, striving to approach the parents’ position with tact, and a struggle between feelings of failure and respect for the parents’ view. The findings indicate that it was crucial to give time, be tactful when meeting parents, and to appear credible and up-to-date. The nurses wanted to be open and respect the parents’ views on vaccination but found it difficult and frustrating to be unable to reach out with their message because their quest was to protect the child.
Ruijs et al. (2012) [42] Netherlands	Grounded theory	To gain insight into the response of healthcare professionals to parents with religious objections to the vaccination of their children.	7 child health clinic doctors, 5 child health clinic nurses and 10 general practitioners	Childhood vaccination	Semi-structured interviews	Three manners of responding to religious objections to vaccination were identified: providing medical information, discussing the decision-making process, and adopting an authoritarian stance. All of the HCPs provided the parents with medical information. In addition, some HCPs discussed the decision-making process. They verified how the decision was made and, if possible, consequences were realized. Sometimes they also discussed religious considerations. Whether the decision-making process was discussed depended on the willingness of the parents to engage in such a discussion and on the religious background, attitudes, and communication skills of the HCPs.
Shahbari et al. (2020) [43] Israel	Qualitative phenomenological research	To examine the impact of trust on the high response rate to vaccinations among the minority Arab population living in Israel.	20 school nurses and nurses working in Family Health Centres (and 70 mothers)	Vaccination against seasonal flu and vaccination against the papilloma virus	Semi-structured interviews	The participants placed the highest trust in the nurses working in the Tipat Halav Family Health Centres run by the Ministry of Health. These nurses are the main communicators of information about childhood vaccinations in Israel. Moreover, the interviewees saw vaccinations as an example of the state offering equal and optimal services to the Arab minority population. In addition, the interviewees consider the explanatory materials to be limited, superficial and not culturally appropriate.
Stretch et al. (2009) [44] UK	Qualitative study	To assess the feasibility and acceptability of providing human papillomavirus vaccination to 12–13-year-olds in 36 schools in Greater Manchester, in the northwest of England.	15 school nurses	Routine human papillomavirus (HPV) vaccination	Semi-structured interviews	School nurses knew how to assess the competency of people under the age of 16 but were still unwilling to vaccinate if parents had refused permission. If parents had not returned the consent form, school nurses were willing to contact parents and also to negotiate with parents who had refused consent. They seemed unaware that parental involvement required the child’s consent to avoid breaking confidentiality. Nurses’ attitudes were influenced by the young appearance and age of the school year group rather than an individual’s level of maturity. They were also confused about the legal guidelines governing consent. School nurses acknowledged the child’s right to vaccination and strongly supported prevention of HPV infection but ultimately believed that it was the parents’ right to give consent. Most were themselves parents and shared other parents’ concerns about the vaccine’s novelty and unknown long-term side effects. Rather than vaccinate without parental consent, school nurses would defer vaccination.

**Table 5 ijerph-18-07584-t005:** Confidence in the Evidence from Reviews of Qualitative research (CERQual) evidence profile.

Summary of Review Findings	Studies Contributing to the Review Findings	Methodological Limitations	Coherence	Relevance	Adequacy of Data	Overall CERQual Assessment of Confidence	Explanation of Decision
Stone hardness—Resistance to vaccination	33; 34; 35; 36; 37; 38; 39; 40; 41; 42; 43	Minor concerns regarding methodological limitations, since there is a lack of clarity regarding the influence of the researcher on the investigation, and vice versa, as well as their possible influence during the data collection and analysis phases	Minor concerns regarding coherence (data very consistent within and across studies)	Moderate concerns regarding relevance, since there are studies in which the health systems are public and others are private, in addition to the fact that the parents who rejected vaccination belonged to very different cultural contexts	Very minor concerns about adequacy of data as the richness of data was generally good. The number of participants and studies is high	High confidence	Minor concerns about coherence and methodological limitations; moderate concerns about relevance; very minor concerns about adequacy of data
Lack of modelling tools—Lack of resources, support and training	34; 35; 38; 40; 41	Minor concerns regarding methodological limitations, since there is a lack of clarity regarding the influence of the researcher on the investigation, and vice versa, as well as their possible influence during the data collection and analysis phases	Minor concerns regarding coherence (data very consistent within and across studies)	Moderate concerns regarding relevance, since there are studies in which the health systems are public and others are private, in addition to the fact that the parents who rejected vaccination belonged to very different cultural contexts	Moderate concerns, as the data were partially rich. Only a few studies in the sample examined in detail the lack of resources and support for healthcare professionals	Moderate confidence	Minor concerns about methodological limitations and coherence; moderate concerns about relevance and adequacy of the data
Rudimentary sculpting—Using personal strategies	33; 34; 38; 39; 40; 41; 42; 43; 44	Minor concerns regarding methodological limitations, since there is a lack of clarity regarding the influence of the researcher on the investigation, and vice versa, as well as their possible influence during the data collection and analysis phases	Minor concerns regarding coherence (data very consistent within and across studies)	Moderate concerns regarding relevance, since there are studies in which the health systems are public and others are private, in addition to the fact that the parents who rejected vaccination belonged to very different cultural contexts	Moderate concerns, as the data were partially rich. Only some primary articles in the sample explained the use of their own strategies by health professionals	Moderate confidence	Minor concerns about methodological limitations and coherence; moderate concerns about relevance and adequacy of the data

## Data Availability

We have published the translation tables of the analysis of the meta-ethnography as Appendix A. Please contact the authors for additional information.

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
