# Peer review of "Encountering Parents Who Are Hesitant or Reluctant to Vaccinate Their Children: A Meta-Ethnography"

_ijerph, 2021, doi:10.3390/ijerph18147584_

Round 1

Reviewer 1 Report

This is an interesting paper which is well-presented in general. The following need to be considered in order for the paper to be in better shape:

  1. Justification is currently poor. Why a meta-ethnography on the subject is necessary? What does a review of qualitative studies has to add to scholarship?
  2. The authors should justify the choice of the specific databases for search.
  3. Inclusion/ exclusion criteria should show in a table and be better justified, including justifying the publication window for the selected studies (ie publishing between 2010 and 2020).
  4. The PRISMA flowchart was followed. However, it would be useful to know the number of studies identified per database.

All the above are essential in order to make sure that the study is reproducible.

Author Response

Thank you for the helpful feedback to improve this important manuscript. We are pleased to submit what we feel is a much-improved version of the manuscript.

In answer to your comments, please see the attached document.

Yours faithfully,

Reviewer 2 Report

The aim of this meta‐ethnography was to synthesise the available body of qualitative work regarding the care experiences of health professionals from community and hospital settings in encounters with parents who are hesitant or reluctant to vaccinate their children.

Nevertheless, there are some issues I would like to point out to enrich the contribution, and the proposal description, which I think must be attended to:

  • First of all, novelty and worth of this work need to be clearly addressed in the abstract and introduction section of the manuscript.
  • Authors must highlight the contribution in a more profound way to identify the analysis of the several sources consulted.
  • More background work need to be incorporated.
  • The manuscript need to be restructured into standard sections viz. introduction, related works, methods, results, discussion and conclusion.
  • There should be separate related work section. To highlight the significant of the meta-analysis and meta-ethnography, the meta-analysis presented by different authors on different aspect of the life need to be briefly summarized in a Table. For this authors may read and refer the following manuscripts:
    • https://doi.org/10.1016/j.compbiomed.2021.104450
    • https://doi.org/10.1007/s11831-020-09412-6
    • 1159/000513733
    • https://doi.org/10.1016/j.ijid.2021.01.013
    • https://doi.org/10.1186/s13012-020-01072-1
    • https://doi.org/10.1371/journal.pone.0245469
  • What are limitation and strength of this work?
  • Finally, there are some grammatical and structural errors that need to be rectified in the revised version of this manuscript.

Author Response

(The authors gave the same response as above.)

Round 2

Reviewer 1 Report

Thank you for considering my feedback. It is in good shape now.